# Photo-thermionic effect in vertical graphene heterostructures

M. Massicotte[1], P. Schmidt[1,*], F. Vialla[1,*], K. Watanabe[2], T. Taniguchi[2], K.J. Tielrooij[1] & F.H.L. Koppens[1,3]

Finding alternative optoelectronic mechanisms that overcome the limitations of conventional semiconductor devices is paramount for detecting and harvesting low-energy photons. A highly promising approach is to drive a current from the thermal energy added to the free-electron bath as a result of light absorption. Successful implementation of this strategy requires a broadband absorber where carriers interact among themselves more strongly than with phonons, as well as energy-selective contacts to extract the excess electronic heat. Here we show that graphene-$WSe_2$-graphene heterostructure devices offer this possibility through the photo-thermionic effect: the absorbed photon energy in graphene is efficiently transferred to the electron bath leading to a thermalized hot carrier distribution. Carriers with energy higher than the Schottky barrier between graphene and $WSe_2$ can be emitted over the barrier, thus creating photocurrent. We experimentally demonstrate that the photo-thermionic effect enables detection of sub-bandgap photons, while being size-scalable, electrically tunable, broadband and ultrafast.

[1] ICFO-Institut de Ciencies Fotoniques, The Barcelona Institute of Science and Technology, Castelldefels, Barcelona 08860, Spain. [2] National Institute for Materials Science, 1-1 Namiki, Tsukuba 305-0044, Japan. [3] ICREA—Institució Catalana de Recerça i Estudis Avancats, Barcelona 08010, Spain. * These authors contributed equally to this work. Correspondence and requests for materials should be addressed to F.H.L.K. (email: frank.koppens@icfo.eu).

Since the discovery of the photoelectric effect in the late nineteenth century[1], a great number of photodetectors that rely on the emission of photoexcited charge carriers have been proposed. These carriers—sometimes referred to as hot carriers, although they are not thermalized with the electron bath—are typically injected over a Schottky barrier between a metal and a semiconductor, allowing detection of photons with energy lower than the semiconductor bandgap (see Fig. 1a). This process, called internal photoemission, has led to the development of visible and near-infrared photodetectors[2,3], which have recently been combined with plasmonic enhancement schemes[4–8]. However, the efficiency of this mechanism drops for photon energy lower than the Schottky barrier height $\Phi_B$ (ref. 9) and is limited by the ability to extract the carriers before they lose their initial energy, which in metals typically occurs on a timescale of $\sim 100$ fs (ref. 10).

A promising way to overcome these limitations is to make use of the excess thermal energy contained in the electron bath. This energy arises from the thermalization of photoexcited carriers with other carriers, which results in a hot carrier distribution with a well-defined temperature $T_e$. For increasing $T_e$, a larger fraction of carriers can overcome the Schottky barrier, creating a current via thermionic emission (Fig. 1b). In this scheme, even photons with energies below $\Phi_B$ can lead to an increase in $T_e$ and subsequently to carrier emission. However, in order to reach high $T_e$, the hot carriers must be weakly coupled to the surrounding phonon bath[11].

Graphene, which has recently emerged as an excellent platform for converting photons into hot carriers[12], has the ideal properties to implement this scheme. Graphene presents strong electron–electron interactions leading to carrier thermalization within $\sim 50$ fs (refs 13,14), where a large fraction ($>50\%$) of the initial energy of photoexcited carriers is transferred to the electronic system[15]. This efficient carrier heating creates a thermalized hot carrier state that is relatively long-lived (longer than 1 ps)[16], owing to weak coupling to the lattice and the environment. These thermalized carriers can thus reach temperatures significantly higher than the phonon bath temperature ($T_e > T_{ph}$) even under continuous-wave (CW) excitation[17] (see Supplementary Note 1 and Supplementary Fig. 1). Moreover, the tunability of the graphene Fermi energy gives control over the height of the Schottky barrier. For these reasons, graphene was recently proposed as a promising material for efficient and tunable thermionic emission of hot carriers[18–20].

Here, we use graphene/WSe$_2$/graphene van der Waals heterostructures to detect low-energy photons (with a wavelength up to 1.5 μm) through photo-thermionic (PTI) emission. Figure 1b shows in detail how the PTI photocurrent is generated: photons are absorbed by graphene, creating electron–hole pairs, which then rapidly equilibrate into a thermalized carrier distribution with an elevated electron temperature $T_e$ compared with the temperature of the lattice $T_{ph}$ and the environment $T_0$; carriers within this distribution with an energy larger than the Schottky barrier height $\Phi_B$ at the graphene/WSe$_2$ interface can be injected into the WSe$_2$ and travel to the other graphene layer. The number of carriers with sufficient energy scales with $e^{\frac{-\phi_B}{k_B T_e}}$, where $k_B$ is the Boltzmann constant.

## Results

### Device structure
In our device, WSe$_2$—a transition metal dichalcogenide with a bandgap $E_g \sim 1.3$ eV—provides an energy barrier between the two graphene sheets with low interfacial defects and reduced Fermi-level pinning. The active device (depicted schematically in Fig. 1c) is encapsulated between layers of hexagonal boron nitride (hBN) which provides a clean,

charge-free environment for the graphene and WSe$_2$ flakes[21]. The device is equipped with an electrostatic bottom gate ($V_G$) that enables control of the Fermi energy $\mu$ and thereby $\Phi_B$ of (mainly) the bottom graphene. All measurements presented in the main text are obtained from one particular device comprising a 28-nm-thick WSe$_2$ flake (see Fig. 1d) and are performed at room temperature with a quasi-CW laser source, unless otherwise mentioned (see Methods). We have studied devices with WSe$_2$ flakes of various thicknesses ($L = 2.2$–40 nm) and obtained similar results, consistent with the PTI effect (see Supplementary Note 2 and Supplementary Figs 2 and 3).

### Photocurrent measurements
The PTI process is driven by the light-induced increase of the thermal energy of the electron gas ($k_B T_e$). Signatures of this mechanism are readily visible in the data presented in Fig. 2. First, the photocurrent spectrum of Fig. 2a shows a sizable, spectrally flat response for photon energies well below the bandgap of WSe$_2$ ($E_{photon} < E_g$). That is expected from a thermally driven photocurrent, given the uniform absorption of graphene in the visible, near-infrared range and the fact that $k_B T_e$ is independent of $E_{photon}$ for constant power[15,22]. Furthermore, the photocurrent generated in this sub-bandgap regime exhibits a striking superlinear dependence on laser power $P$ (Fig. 2b,c). This is a direct consequence of the thermal activation of carriers over the Schottky barrier, which, in first approximation, scales exponentially with $P$ (see Methods). In contrast, the photocurrent in the above-bandgap regime ($E_{photon} > E_g$) varies strongly with $E_{photon}$ and scales linearly with $P$. This photoresponse is characteristic of light absorption in WSe$_2$ and transfer of photoexcited carriers to the graphene electrodes, a process driven by the potential drop across the WSe$_2$ layer[23–25].

Alternative photocurrent generation mechanisms are less likely to contribute to the observed sub-bandgap photocurrent. To verify this, we measured a device with a Au/WSe$_2$ interface, where photocurrent is generated by internal photoemission of non-thermalized photoexcited carriers (see Supplementary Note 3 and Supplementary Fig. 4). This device shows a strong dependence on $E_{photon}$ along with a cutoff energy at $E_{photon} = \Phi_B$, and a linear power dependence—clearly at odds with our observations for G/WSe$_2$/G (where G stands for graphene) devices. We note that multi-photon absorption followed by charge transfer could also lead to a superlinear power dependence, but the laser intensity required to induce significant two-photon absorption in either graphene or WSe$_2$ is at least 1–2 orders of magnitude higher than the one used in our experiment (smaller than 1 GW cm$^{-2}$) (refs 26,27). Similarly, the photo-thermoelectric and bolometric effects could generate sub-bandgap photocurrent, but both would have a sublinear—rather than the observed superlinear—power dependence[16,28].

To further verify that the sub-bandgap photocurrent stems from the PTI effect, we perform time-resolved photocurrent measurements by varying the time delay $\Delta t$ between two sub-picosecond laser pulses generated by a Ti:sapphire laser (see Supplementary Note 4 and Supplementary Fig. 5). From the dynamics of the positive correlation signal (due to the superlinear power dependence) in Fig. 2d, we extract a characteristic decay time $\tau_{cool}$ of 1.3 ps, which is on the order of the cooling time of hot carriers in graphene[16,22]. All together the observations presented in Fig. 2 suggest that the sub-bandgap, superlinear, picosecond photocurrent is governed by the PTI effect.

### Electrical tuning of the PTI effect
In contrast to bulk metal–semiconductor systems, this graphene-based heterostructure offers the possibility to tune the Schottky barrier, and therefore the magnitude of the PTI photocurrent, using the interlayer bias

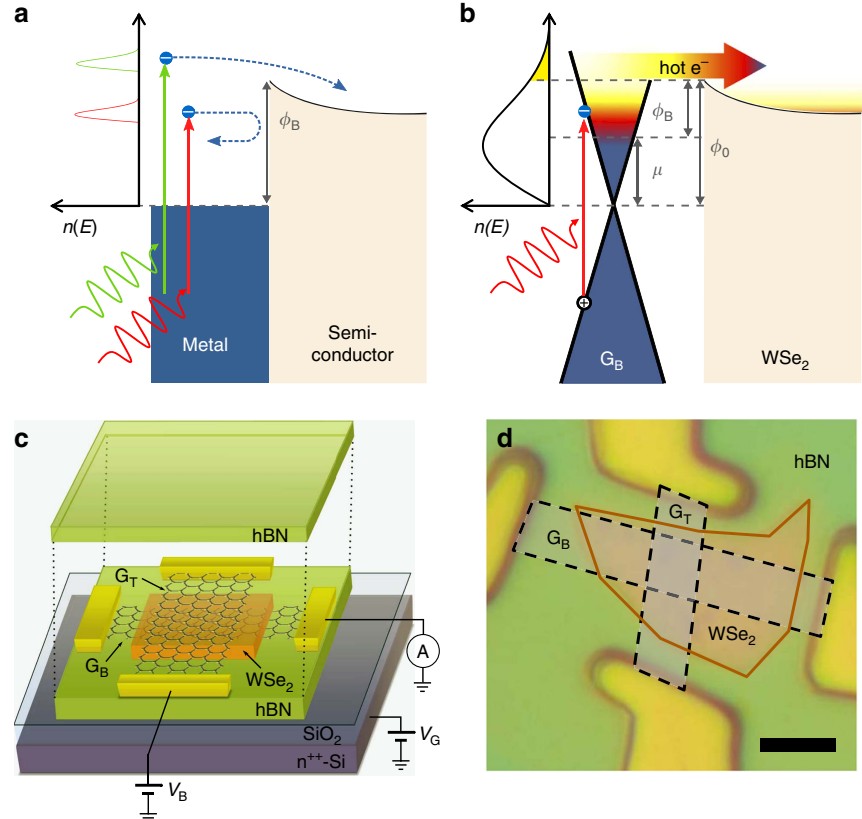

**Figure 1 | The photo-thermionic effect and device structure.** (**a**) Simplified band diagram illustrating the internal photoemission process taking place at a metal–semiconductor interface. Non-thermalized photoexcited carriers in metal with sufficient energy to overcome the Schottky barrier $\Phi_B$ can be injected into the semiconductor before they lose their initial energy (within 100 fs for conventional metals[10]). The portion of the energy band filled by electrons and the bandgap of the semiconductor are shaded in blue and pale orange, respectively. Low (high) energy photon and the electronic transition following their absorption are represented by red (green) sinusoidal and vertical arrows. The out-of-equilibrium electron distributions $n(E)$ resulting from these processes are illustrated on the left-hand side with the corresponding colours. Photoexcited electrons are depicted by blue dots and their possible transfer path is represented by blue dashed arrows. (**b**) Simplified band diagram of the PTI effect at a G/WSe$_2$ interface. The ultrafast thermalization of photoexcited carriers in graphene gives rise to a hot-electron distribution $n(E)$ with a lifetime longer than 1 ps. As the number of electrons in the hot tail (yellow shaded area) of $n(E)$ increases, more electrons are emitted over the Schottky barrier $\Phi_B$, which generates a larger thermionic current (represented by the horizontal arrow). The colour gradient from blue to yellow illustrates the heat contained in the electron distribution. The offset between the graphene neutrality point and WSe$_2$ conduction edge is denoted by $\Phi_0$ and was experimentally determined to be 0.54 eV (ref. 28). (**c**) Schematic representation of the heterostructure on a 285-nm-thick SiO$_2$/Si substrate, to which a gate voltage ($V_G$) is applied to modify the Fermi-level $\mu$ of the bottom graphene. An interlayer bias voltage ($V_B$) between the top (G$_T$) and bottom (G$_B$) graphene flakes can be applied, and current or photocurrent flowing through G$_B$ is measured. (**d**) Optical image of a heterostructure composed of a 28-nm-thick WSe$_2$ flake. The top and bottom hBN flakes are 10 and 70 nm thick, respectively. For clarity, graphene flakes are shaded in grey and outlined by a black dashed line, whereas WSe$_2$ is coloured in orange and outlined by an orange line. Scale bar, 5 μm.

voltage ($V_B$) and gate voltage ($V_G$). Applying these voltages is necessary in order to generate a finite photocurrent, as it breaks the symmetry of our device, which is composed of two G/WSe$_2$ Schottky barriers with opposite polarity. As the infrared photo-current maps ($E_{photon} = 0.8$ eV) in Fig. 3a,b indicate, the inter-layer voltage $V_B$ essentially controls over which of the two Schottky barriers hot carriers are injected: for $V_B = -0.6$ V (Fig. 3a), the photoactive region corresponds to the area where the top graphene overlaps with the WSe$_2$ layer (G$_T$/WSe$_2$), whereas the interface with the bottom graphene (G$_B$/WSe$_2$) is photoactive for $V_B = +0.6$ V (Fig. 3b). In Fig. 3c,d, we examine the photocurrent originating from regions containing a single G/WSe$_2$ interface, thus allowing us to assess each Schottky barrier individually. To create a current, hot carriers need to be emitted over the G/WSe$_2$ interface and subsequently transported along the WSe$_2$ layer and collected by the other graphene electrode, as illustrated in the insets of Fig. 3c,d. When the interlayer bias $V_B$ makes this process energetically favourable, each Schottky barrier

gives rise to a photocurrent with a specific sign. The photocurrent generated in the G/WSe$_2$/G region (Fig. 3e) exhibits both signs as it stems from charge injection over both top and bottom Schottky barriers. From the photocurrent sign associated with each layer, we deduce that hot electrons, rather than holes, are pre-dominantly emitted over both Schottky barriers. This is expected given the work functions of graphene and electron affinity of WSe$_2$ (ref. 29).

One of the hallmarks of thermionic emission is its exponential dependence on the Schottky barrier height. In our device, the gate voltage $V_G$ provides a crucial way of enhancing the photocurrent by controlling the height of the G$_B$/WSe$_2$ Schottky barrier via the tuning of the Fermi energy of G$_B$. As Fig. 3f demonstrates, doping the bottom graphene layer with electrons by applying a positive gate voltage $V_G$ effectively lowers $\Phi_B$ and results in a strong increase in photocurrent. At high $V_G$ (low $\Phi_B$), the device reaches a responsivity of up to 0.12 mAW$^{-1}$ at wavelength $\lambda = 1,500$ nm, which, for 0.5% light absorption in graphene[30], translates into an

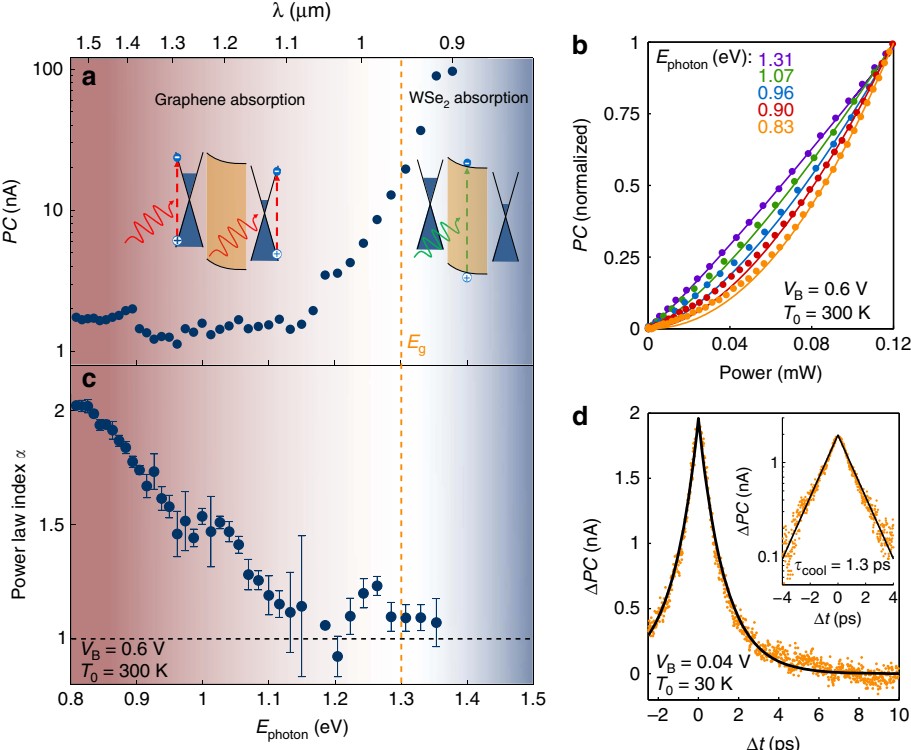

**Figure 2 | Experimental signatures of photo-thermionic emission.** (**a**) Photocurrent (PC) spectrum measured at room temperature in the G/WSe₂/G region with laser power $P = 90\,\mu W$, $V_B = 0.6\,V$ and $V_G = 0\,V$ (same conditions for **b,c**). The insets illustrate the absorption process taking place in the different photoresponse regimes: absorption in WSe₂ for $E_{photon} > E_g$ and absorption in graphene for $E_{photon} < E_g$. The transition between these two regimes is represented by the background colour gradient, where red (blue) corresponds to the graphene (WSe₂) absorption regimes. The vertical orange dashed line corresponds to the energy of the bulk WSe₂ bandgap. (**b**) Power dependence of the photocurrent for various values of photon energy $E_{photon}$. The dots represent experimental data and the solid lines are power law fits ($PC \propto P^\alpha$) obtained with a fit range $P = 70$–$120\,\mu W$. (**c**) Fitted power law index $\alpha$ versus photon energy, showing the transition from linear to superlinear power dependence. This transition occurs around $E_{photon} = E_g$, the indirect bandgap of WSe₂. The error bars correspond to the s.d. obtained from the linear fit. (**d**) Time-resolved photocurrent change $\Delta PC(\Delta t) = PC(\Delta t) - PC(\Delta t \to \infty)$, measured using the setup and technique described in ref. 24 with an average laser power of $260\,\mu W$ (wavelength 800 nm), at low temperature (30 K) and bias ($V_B = 0.04\,V$) in order to suppress the contribution of the photocurrent originating from WSe₂ absorption. Experimental data are represented by orange dots and the solid black line is a decaying exponential fit with time constant $\tau_{cool} = 1.3\,ps$. Inset: same data and fit in logarithmic scale.

internal quantum efficiency (IQE) of 2%. These figures of merit are similar to those obtained in devices using the in-plane photo-thermoelectric effect[31] and can be further improved by adjusting the relevant physical parameters as discussed below.

**Theoretical model of the PTI effect.** The gate tunability of the PTI process and its distinct power dependence allow for a quantitative comparison of our measurements with a Schottky barrier model based on the Landauer transport formalism[32] (see Methods). In this model, the photocurrent depends on the fraction of carriers with enough energy to overcome the barrier, governed by $T_e$ and $\Phi_B$, and on the carrier injection time $\tau_{inj}$. The values for $\Phi_B(V_G)$ are extracted from temperature-dependent dark current measurements (Fig. 4a) and are consistent with a band offset $\Phi_0$ of 0.54 eV (ref. 29). For simplicity, we assume that heat in the electronic system dissipates through a single, rate-limiting cooling pathway characterized by a thermal conductance $\Gamma$, such that under steady-state conditions the increase in temperature is proportional to $P/\Gamma$ (see Supplementary Note 1).

Figure 4b compares the measured and fitted PC as a function of $\Phi_B(V_G)$ and laser power. This two-dimensional fit yields a carrier injection time $\tau_{inj} = 47 \pm 10\,ps$ and a thermal conductance $\Gamma = 0.5 \pm 0.3\,MWm^{-2}K^{-1}$. This value of $\tau_{inj}$ is almost identical to the one found for ideal G/Si Schottky barriers[32], while the one obtained for $\Gamma$ matches the predicted thermal conductance of

G/hBN interfaces due to electron coupling with SPP phonons[33] and is also consistent with disorder-enhanced supercollisions with acoustic phonons[16] (see Supplementary Note 1). The excellent agreement between model and experiment is clearly visible in Fig. 4c,d. We note that the same measurements were performed at other ambient temperatures ($T_0 = 230$ and $330\,K$) and the analysis yields very similar results (see Supplementary Note 5 and Supplementary Fig. 6).

**Discussion**

The device modelling and extracted physical parameters provide important insights into how to improve the efficiency of the PTI process. They also explain why this mechanism dominates the photoresponse of graphene/semiconductor heterostructures, while being absent for metal/semiconductor devices. The reason is that the thermal conductance $\Gamma$ of our graphene-based device is more than 2 orders of magnitude smaller than the thermal conductance due to electron–phonon coupling in thin ($\sim 10\,nm$) metal films[10] (see Supplementary Note 3). Hence, thermalized hot carriers in metals do not reach a sufficiently high temperature to generate significant PTI photocurrent. Strategies to substantially increase the device efficiency include further reduction of the thermal conductance in graphene-based devices, for example, by using a nonpolar encapsulating material[33]. Likewise, the efficiency of the process can be readily improved by lowering

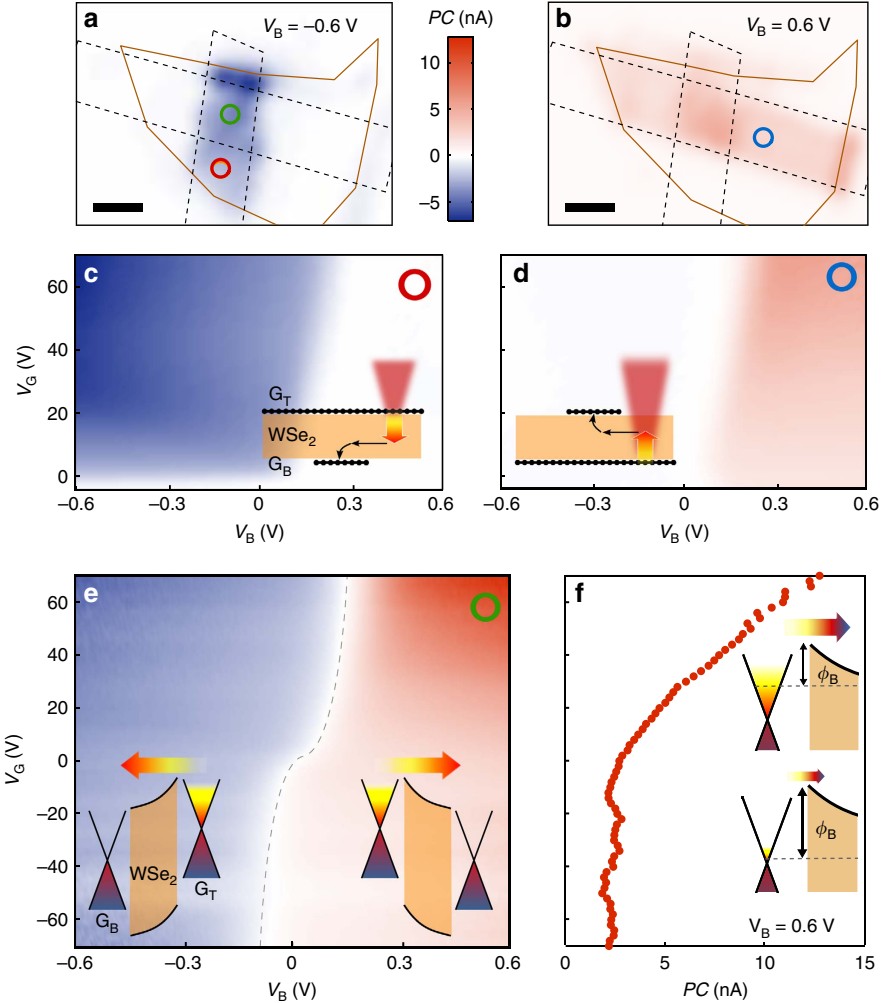

**Figure 3 | Tunable photo-thermionic response. (a,b)** $PC$ maps of the device shown in Fig. 1d measured with an interlayer bias voltage $V_B$ of **(a)** $-0.6$ V and **(b)** 0.6 V, and $V_G = 0$ V. The graphene flakes are outlined by black dotted lines and the WSe₂ flake by solid orange lines, as in Fig. 1d. Scale bar, 3 μm. **(c–e)** $PC$ versus $V_B$ and $V_G$ measured on single Schottky barriers formed by **(c)** top or **(d)** bottom graphene and WSe₂, as well as **(e)** double G/WSe₂/G interfaces. The coloured circle (green, red and blue) in the upper right corner of each measurement corresponds to the position of the focused laser beam which are indicated on the $PC$ maps **(a,b)**. The black dashed line in **e** indicates where $PC$ is null. All measurements are scaled to the same colour bar. Insets of **c,d**: side view of the heterostructure illustrating the generation and transport of hot carriers from one graphene flake to the other. Insets of **e**: band diagrams depicting the PTI effect in G/WSe₂/G for $V_B < 0$ (left) and $V_B > 0$ (right). **(f)** $PC$ versus $V_G$ taken from **(e)** at $V_B = 0.6$ V. Inset: band diagrams of the G_B/WSe₂ Schottky barrier at low (bottom) and high (top) $V_G$ illustrating the increase of PTI emission resulting from the lowering of $\Phi_B$. All measurements are performed at room temperature, with $E_{photon} = 0.8$ eV and $P = 110$ μW.

$\Phi_B$. Indeed, we find that the PTI efficiency increases by one order of magnitude (up to 20%) by extrapolating the IQE to higher $T_e$ ($\sim 1,000$ K) or lower $\Phi_B$ ($\sim 0.06$ eV, see Supplementary Note 6 and Supplementary Fig. 7). Moreover, our model suggests that the efficiency can be greatly enhanced by reducing the carrier injection time $\tau_{inj}$, which is related to the coupling energy between adjacent layers. The long $\tau_{inj}$ obtained from our fit appears to be one of the main factors limiting the observed IQE and is presumably due to momentum mismatch between electronic states in the two adjacent materials. The interlayer transfer of charge carriers and heat in van der Waals heterostructures is currently not well understood and further studies are needed in order to unveil the limits of the PTI efficiency.

We finally note that the PTI effect shows some similarities to photon-enhanced thermionic emission (PETE), with the important differences that for PETE the photoexcited carriers are in thermal equilibrium with the lattice of a hot semiconductor and are emitted over a vacuum energy barrier[34,35]. There are also important resemblances between the PTI mechanism and the concept of hot carrier solar cells, since both require decoupling of the electron and phonon baths and energy-selective contacts[11,36]. Both PETE devices and hot-solar cells have an interesting potential for power conversion, but harvesting low-energy photons is limited by the bandgap of the semiconducting absorber. Interestingly, in our PTI device, which has a very simple geometry and operates at room temperature, we also find a gate-dependent open-circuit voltage (of the photocurrent) of up to 0.17 V with a fill factor of 38%. This effect, observable in Fig. 3e, opens up a promising avenue for infrared energy harvesting using graphene as the active material[37]. Furthermore, the PTI mechanism should work over an extremely broad wavelength range, including the mid-infrared and far-infrared (terahertz) regions and can be used for ultrafast photodetection, given that the signal recovery time is on the order of picoseconds. Finally, these vertical thermionic devices have a scalable active area and can be easily integrated with conventional and flexible solid-state devices. These features make

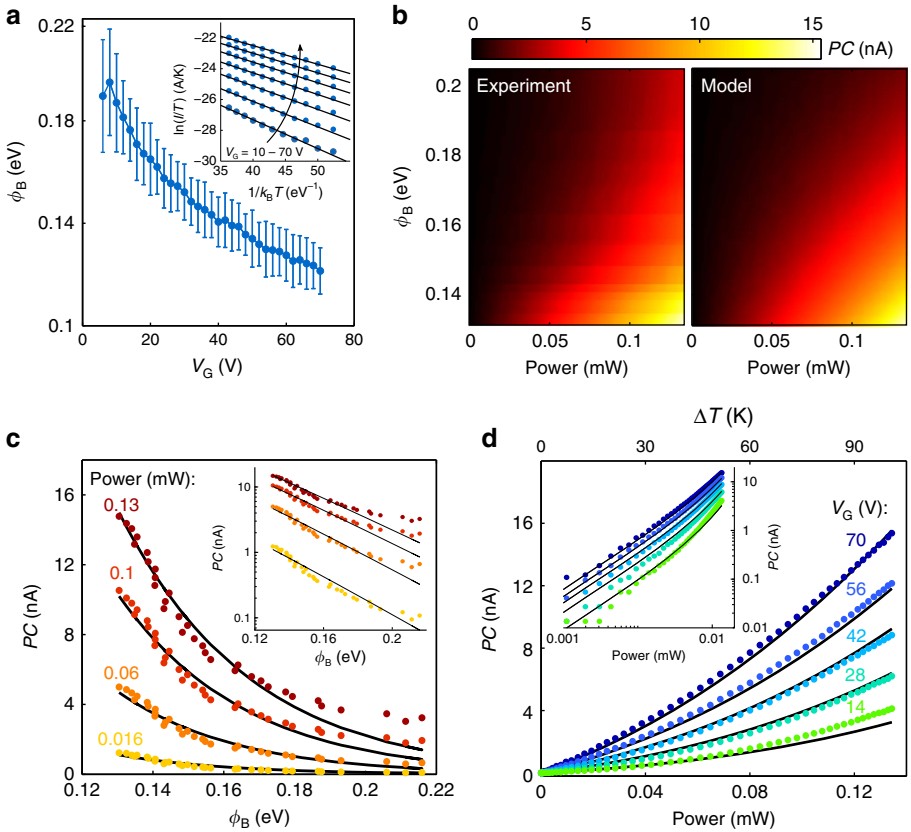

**Figure 4 | Comparison between data and photo-thermionic model. (a)** Schottky barrier height $\Phi_B$ versus $V_G$ extracted from the temperature-dependence of dark current measurements. Inset: Arrhenius plot of the dark current at different $V_G$ and $V_B = 0.36$ V. Experimental data are represented by blue dots and the solid black lines are linear fits. The error bars in the main panel correspond to the standard deviation obtained from these fits. **(b)** $PC$ versus $\Phi_B$ and laser power $P$, measured (left plot) and according to our PTI model (right plot). $PC$ is measured at room temperature, with $E_{photon} = 0.8$ eV and $V_B = 0.36$ V. **(c)** $PC$ versus $\Phi_B$ for different values of $P$ and **(d)** $PC$ versus $P$ for various values of $\Phi_B$ taken from **b**. The data points correspond to the experiment and the solid black lines to the model. The upper horizontal axis shows the rise in electronic temperature $\Delta T = T_e - T_0$ (extracted from the fit of the model to the experiment). Insets of **c,d**: same experimental data and theoretical curves in logarithmic scale.

the photo-thermionic effect a highly promising mechanism for a plethora of optoelectronic applications[38].

## Methods

**Device fabrication and optoelectronics measurements.** The heterostructures are fabricated the same way as described in ref. 24. Photocurrent is generated by focusing a supercontinuum laser (NKT Photonics SuperK extreme, repetition rate $f = 40$ MHz and pulse duration $dt = 100$ ps) with a microscope objective (Olympus LCPlanN × 50) on the device. The photocurrent is measured using a preamplifier and a lock-in amplifier synchronized with a mechanical chopper at 117 Hz.

**PTI emission model.** In the reverse-bias regime, the Schottky barrier model based on the Landauer transport formalism[32] (see Supplementary Note 7) predicts that the current density $J$ thermionically emitted over a Schottky barrier of height $\Phi_B$ at temperature $T$ is

$$J(T) = \frac{2}{\pi} \frac{e_0}{\tau_{inj}} \left(\frac{k_B T}{\hbar v_F}\right)^2 \left(\frac{\Phi_0}{k_B T} + 1\right) exp\left(\frac{-\Phi_B}{k_B T}\right), \qquad (1)$$

where $e_0$ is the elementary charge, $k_B$ is the Boltzmann's constant, $\hbar$ is the reduced Planck's constant, $v_F$ is the graphene Fermi velocity, $\Phi_0$ is the band offset at the G/WSe$_2$ interface (0.54 eV) and $\tau_{inj}$ is the charge injection time. In our experiment, the thermionic photocurrent ($PC$) we measure is produced by the increase of electronic temperature $\Delta T = T_e - T_0$ upon illumination of the device with a quasi-CW laser at $\lambda = 1,500$ nm. Hence, it follows that the photocurrent is $PC = AD \left[J(T_0 + \Delta T) - J(T_0)\right]$, where $A$ is the area of the laser beam (laser spot size of 1.75 μm), $D$ is the duty cycle ($D = dt \cdot f = 0.04\%$) and $T_0$ is the ambient temperature. Using these equations and assuming $T_0 \gg \Delta T$, one can show that $PC \propto \Delta T + \frac{\phi_B}{2k_B T_0^2} \Delta T^2 + \dots$, which makes evident the superlinear behaviour of the photocurrent. Finally, we assume that the rise in electronic temperature created by each pulse is $\Delta T = \alpha \eta_{heat} P/AD\Gamma$, where $\alpha$ is the light absorption in graphene

(0.5%), $\eta_{heat}$ is the fraction of absorbed energy that is transferred to the electron bath ($\sim 70\%$) (ref. 15), $P$ is the average power of the laser and $\Gamma$ is the thermal conductance of the rate-limiting thermal dissipation step (see Supplementary Note 1).

**Data availability.** The data that support the findings of this study are available from the corresponding author upon request.

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

## Acknowledgements

We are grateful to Qiong Ma, Pablo Jarillo-Herrero, Mark Lundeberg and Ilya Goykhman for valuable discussions. M.M. thanks the Natural Sciences and Engineering Research Council of Canada (PGSD3-426325-2012). P.S. acknowledges financial support by a scholarship from the 'la Caixa' Banking Foundation. F.V. acknowledges the financial support from Marie-Curie International Fellowship COFUND and ICFOnest program. K.J.T. acknowledges the financial support from Mineco (FIS2014-59639-JIN). F.H.L.K. acknowledges support by Fundacio Cellex Barcelona, the ERC Career integration grant (294056, GRANOP), the ERC starting grant (307806, CarbonLight), the Mineco grants RYC-2012-12281 and FIS2013-47161-P and support by the EC under the Graphene Flagship (contract no. CNECT-ICT-604391).

## Author contributions

M.M. and F.H.L.K. conceived and designed the experiments. M.M., P.S. and F.V. fabricated the samples, carried out the experiments and M.M. performed the data analysis. K.W. and T.T provided boron nitride crystals. M.M., F.V., K.J.T., P.S. and F.H.L.K discussed the results and co-wrote the manuscript.

## Additional information

**Competing financial interests:** The authors declare no competing financial interests.

**How to cite this article**: Massicotte, M. *et al.* Photo-thermionic effect in vertical graphene heterostructures. *Nat. Commun.* 7:12174 doi: 10.1038/ncomms12174 (2016).

