## [Peer review file · Nature Communications]

Reviewers' comments:

Reviewer #1 (Remarks to the Author):

The paper reports a photon-enhanced thermionic effect (termed the photo-thermionic effect) in graphene-WSe₂-graphene heterostructures. The work is an extension of that reported recently (ref 23) by the same group in which they demonstrated picosecond photo-response in the same heterostructure device used in this study. While the previous work focused on extraction of the photo-excited carriers in the WSe₂ layer by top and bottom graphene electrodes, the present work focuses on emission of thermalized hot electrons in graphene over a Schottky barrier at the graphene-WSe₂ interface. As highlighted in the supplementary section, previous research has reported strong carrier-carrier scattering in graphene that produces a thermalized photo-generated carrier population hotter than the lattice. Carriers with energy higher than the Schottky barrier are then emitted over the barrier, thus resulting in photocurrent. The process allows detection of sub-band gap photons. The supplementary section (iii) explains why internal photo emission in graphene is suppressed and the observed photocurrent is indeed due to a photon-enhanced thermionic effect. Photon-enhanced thermionic current from a semiconductor cathode to metallic anode has been proposed previously [see Schwede JW, Bargatin I, Riley DC, Hardin BE, Rosenthal SJ, Sun Y, et al. Photon-enhanced thermionic emission for solar concentrator systems. *Nat Mater.* 2010;9(9):762-7] for concentrated solar concentrator (this prior work and related subsequent studies deserve attribution in the present manuscript), whereas the present work demonstrates the concept for a graphene-WSe₂ heterostructure. Some relevant unanswered issues follow:

1. What is the need for encapsulating the graphene-WSe₂-graphene heterostructure between stacks of h-BN? What role does h-BN play in the overall experimental scheme?
2. The electrochemical potential and Schottky barrier height should be quantified in the schematic of Figure 1.

The work in general is certainly of high academic interest, but likely more so to the specialist community that is working on exploiting hot electron effects in real devices.

Reviewer #2 (Remarks to the Author):

The authors report on a novel device architecture for photodetection utilizing 2D-materials and utilization of a previously not explored physical effect (PTI) in 2D-materials for the generation of a photocurrent. A three-layer stack of 2D-materials (Graphene and WSe₂) is used to form a Schottky-junction whose barrier height can be controlled by application of a gate voltage. Upon light excitation, electron-hole pairs created in graphene raise the electronic temperature of the electron bath and hot carriers can overcome the barrier and create a photocurrent.

To the reviewer's knowledge, this effect has not been reported in 2D-materials for photodetection, so far. Presented results are of great interest, not only to the graphene/2D-material community, but also to a wider audience. (Potential) and reported quantum efficiencies make this device architecture not only interesting for photodetection but also for PV applications. Further, it can be used over a wide wavelength range at fast time scales.

Measurements seem to have been carried out with a high standard and data has been analyzed and presented in a way that is up to the standard of Nat. Comms.

Conclusions drawn from the experiments are sound & robust and have been reproduced on several devices. Data is backed-up by theoretical calculations, reproducing the experimental data.

Overall, the paper is very well written and requires only minor corrections/additions before it can be accepted for publication:

- In Fig. 1 a,b the authors depict the band diagram of the system. The authors seem to depict the Schottky junction formation only at one of the interfaces. However, since it is a symmetric system I would assume the Schottky junction to form at both graphene/WSe₂ interfaces. Can the authors please comment what impact two junctions will have on device operation, e.g. once the electron is excited over the first graphene/WSe₂ Schottky barrier it will travel through the WSe₂ until it reaches the second Schottky barrier. Does this reduce the efficiency of the device? Would it be favourable to make the device asymmetric, e.g. use a different material as bottom electrode than graphene?

- p.5, "Electrical tuning of the PTI effect". WSe₂ is a semiconductor, as such it is called electron affinity not work function.

Best wishes.

Reviewer #3 (Remarks to the Author):

(a) Well documented arguments through carefully arranged references

(b) I would recommend little more theory regarding the Landauer formula in the main paper; yes the authors correctly provide specific references (dealing with ideal graphene-Schottky junctions) where the formula is "used" for matching experimental results; I think it would be helpful to readers involved in graphene-based devices to at least "see" a discussion of Landauer's approach and offer a brief description of the formula and its components, especially the role of Graphene's DOS in relation with the DOS of the adjacent layers in the proposed device.

Overall I find the paper to be a well-written experimental paper with a very interesting conclusion: that even sub-bandgap photons may push thermionically-escaping electrons to overcome the Schottky barrier.

This is a very interesting conclusion (well documented through experiment) triggering interest in further theoretical investigations.

We thank all referees for their constructive and generally positive comments on our manuscript.

We respond to the reviewers' comments below.

Please note that reviewers' comments are in red font and that all changes made to the manuscript and Supplementary information are highlighted in yellow.

Reviewer #1 (Remarks to the Author):

The paper reports a photon-enhanced thermionic effect (termed the photo-thermionic effect) in graphene-WSe₂-graphene heterostructures. The work is an extension of that reported recently (ref 23) by the same group in which they demonstrated picosecond photo-response in the same heterostructure device used in this study. While the previous work focused on extraction of the photo-excited carriers in the WSe₂ layer by top and bottom graphene electrodes, the present work focuses on emission of thermalized hot electrons in graphene over a Schottky barrier at the graphene-WSe₂ interface. As highlighted in the supplementary section, previous research has reported strong carrier-carrier scattering in graphene that produces a thermalized photo-generated carrier population hotter than the lattice. Carriers with energy higher than the Schottky barrier are then emitted over the barrier, thus resulting in photocurrent. The process allows detection of sub-band gap photons. The supplementary section (iii) explains why internal photo emission in graphene is suppressed and the observed photocurrent is indeed due to a photon-enhanced thermionic effect. Photon-enhanced thermionic current from a semiconductor cathode to metallic anode has been proposed previously [see Schwede JW, Bargatin I, Riley DC, Hardin BE, Rosenthal SJ, Sun Y, et al. Photon-enhanced thermionic emission for solar concentrator systems. *Nat Mater.* 2010;9(9):762-7] for concentrated solar concentrator (this prior work and related subsequent studies deserve attribution in the present manuscript), whereas the present work demonstrates the concept for a graphene-WSe₂ heterostructure.

We thank the reviewer for his/her thoughtful and constructive feedback. As the reviewer correctly points out, the photo-thermionic (PTI) effect that we report in graphene-WSe₂ heterostructure shares some similarities with photon-enhanced thermionic emission (PETE) previously demonstrated. In the discussion section of the originally submitted manuscript (p. 8), we actually already explicitly mention this resemblance to PETE, as well as the hot-carrier solar cell concept (which was proposed before PETE), including a reference to the paper by Schwede *et al.* [1], as indicated by the reviewer. We also specify important differences between PTI and PETE. In the latter case (PETE), hot electrons are emitted from a semiconductor to a metal over a vacuum energy barrier. This vacuum energy barrier is not always practical in devices. In our work, through the PTI process, hot carriers are emitted from a unique metal-like material (graphene) to a semiconductor over a solid-state Schottky barrier. Most importantly, in PETE devices, photoexcited carriers are in thermal equilibrium with the lattice of the semiconductor while for the mechanism we report (PTI), hot carriers are thermalized among themselves but not with the lattice, owing to the weak electron-phonon coupling of graphene (see Supplementary section 1). Nevertheless, we agree with the reviewer that more studies on PETE deserve to be highlighted, so the new manuscript now includes a citation of the subsequent work of Schwede *et al.* [2] (page 8).

We stress that the mechanism we introduce is fundamentally new, enabled by unique material properties that were not available before, and furthermore useful for detection and harvesting of low energy photons, which has potential applications in a broad range of applications, such as broadband photodetection and infrared light harvesting.

Some relevant unanswered issues follow:

1. What is the need for encapsulating the graphene-WSe₂-graphene heterostructure between stacks of h-BN? What role does h-BN play in the overall experimental scheme?

Hexagonal Boron Nitride (h-BN) is a wide bandgap (~6 eV) insulator and therefore, it is optically and electrically inactive in our optoelectronic devices. It mainly serves two functions: First, the top h-BN flakes help to pick up graphene and WSe₂ flakes during the layer assembly step, as first described by Wang et al. [3]. Second, the two h-BN flakes act as encapsulation layers and provide a clean, stable environment (atomically flat substrate, free of dangling bonds) for the graphene flakes with little residual doping. This facilitates the tuning of the graphene Fermi level via electrostatic gating. To address this unanswered issue, we added a short description of the role of the h-BN in the new manuscript (Page 3).

2. The electrochemical potential and Schottky barrier height should be quantified in the schematic of Figure 1.

We thank the referee for this useful suggestion. One of the main advantages of using graphene as a hot carrier emitter is the tunability of its electrochemical potential (μ), and therefore the height of the Schottky barrier (Φ_B). Consequently, μ and Φ_B are not restricted to a single value but can be tuned over a certain energy range. However, the band offset Φ_0 , i.e. the energy difference between the bottom of the conduction band and the Dirac point of graphene ($\mu + \Phi_B$), remains constant. In the new version of Figure 1 we added a symbol to represent Φ_0 and quantify it in the caption.

- [1] Schwede, J. W., Bargatin, I., Riley, D. C., Hardin, B. E., Rosenthal, S. J., Sun, Y., ... Melosh, N. a. (2010). Photon-enhanced thermionic emission for solar concentrator systems. *Nature Materials*, 9(9), 762–7.
- [2] Schwede, J. W., Sarmiento, T., Narasimhan, V. K., Rosenthal, S. J., Riley, D. C., Schmitt, F., ... Shen, Z.-X. (2013). Photon-enhanced thermionic emission from heterostructures with low interface recombination. *Nature Communications*, 4, 1576.
- [3] Wang, L., Meric, I., Huang, P. Y., Gao, Q., Gao, Y., Tran, H., ... Dean, C. R. (2013). One-dimensional electrical contact to a two-dimensional material. *Science (New York, N.Y.)*, 342(6158), 614–7.

Reviewer #2 (Remarks to the Author):

The authors report on a novel device architecture for photodetection utilizing 2D-materials and utilization of a previously not explored physical effect (PTI) in 2D-materials for the generation of a photocurrent. A three-layer stack of 2D-materials (Graphene and WSe₂) is used to form a Schottky-junction whose barrier height can be controlled by application of a gate voltage. Upon light excitation, electron-hole pairs created in graphene raise the electronic temperature of the electron bath and hot carriers can overcome the barrier and create a photocurrent.

To the reviewer's knowledge, this effect has not been reported in 2D-materials for photodetection, so far. Presented results are of great interest, not only to the graphene/2D-material community, but also to a wider audience. (Potential) and reported quantum efficiencies make this device architecture not only interesting for photodetection but also for PV applications. Further, it can be used over a wide wavelength range at fast time scales.

Measurements seem to have been carried out with a high standard and data has been analyzed and presented in a way that is up to the standard of Nat. Comms.

Conclusions drawn from the experiments are sound & robust and have been reproduced on several devices. Data is backed-up by theoretical calculations, reproducing the experimental data.

Overall, the paper is very well written and requires only minor corrections/additions before it can be accepted for publication:

- In Fig. 1 a,b the authors depict the band diagram of the system. The authors seem to depict the Schottky junction formation only at one of the interfaces. However, since it is a symmetric system I would assume the Schottky junction to form at both graphene/WSe₂ interfaces. Can the authors please comment what impact two junctions will have on device operation, e.g. once the electron is excited over the first graphene/WSe₂ Schottky barrier it will travel through the WSe₂ until it reaches the second Schottky barrier. Does this reduce the efficiency of the device? Would it be favourable to make the device asymmetric, e.g. use a different material as bottom electrode than graphene?

- p.5, "Electrical tuning of the PTI effect". WSe₂ is a semiconductor, as such it is called electron affinity not work function.

We first thank the referee for the very positive review and his/her two relevant remarks, which we address here:

1- The referee rightfully points out that our device contains two Schottky junctions, i.e. one at each graphene/WSe₂ interface. And indeed, in order to generate a finite photocurrent and increase the efficiency, the symmetry of the system needs to be broken. As we explain in the section on “*Electrical tuning of the PTI effect*”, we achieve this in our devices by applying either 1) a bias voltage V_B between the two graphene sheets or 2) a gate voltage V_G which mainly tunes the Fermi level of the bottom graphene layer. The black dotted line in Figure 3e) indicates the value of V_B and V_G at which the system is symmetric and therefore the photocurrent is zero. The highest photocurrent is reached when the system is made as asymmetric as possible (high V_B and V_G). In this regime (reversed bias), the overall efficiency of the photocurrent is limited only by the one Schottky barrier where hot electrons are injected. Alternatively, as the referee suggests, the system could also be made asymmetric by using a material different than graphene for the ‘cold’ anode. This material should have a different band offset (or Schottky barrier height) with WSe₂ than that of G/WSe₂ in order to break the symmetry. We now mention more explicitly the requirement of symmetry breaking to have a nonzero photocurrent (Page 5).

2- We thank the referee for this observation. We changed this sentence to : “This is expected given the work functions of graphene and electron affinity of WSe₂” (Page 6)

Reviewer #3 (Remarks to the Author):

- (a) Well documented arguments through carefully arranged references
- (b) I would recommend little more theory regarding the Landauer formula in the main paper; yes the authors correctly provide specific references (dealing with ideal graphene-Schottky junctions) where the formula is "used" for matching experimental results; I think it would be helpful to readers involved in graphene-based devices to at least "see" a discussion of Landauer's approach and offer a brief description of the formula and its components, especially the role of Graphene's DOS in relation with the DOS of the adjacent layers in the proposed device.

Overall I find the paper to be a well-written experimental paper with a very interesting conclusion: that even sub-bandgap photons may push thermionically-escaping electrons to overcome the Schottky barrier.

This is a very interesting conclusion (well documented through experiment) triggering interest in further theoretical investigations.

We are grateful to the referee for the positive evaluation of our paper and for the constructive suggestions. The referee recommends adding more theoretical details regarding the Landauer equation used in our PTI model. Although this model has already been documented in previous publications, we acknowledge that more details would help the reader understand better the model and its underlying assumptions. For this reason, we added a new section (section VII) in the Supplementary Information where we explain the origin of this equation in more depth.

REVIEWERS' COMMENTS:

Reviewer #1 (Remarks to the Author):

The response to the review satisfactorily addresses the major points raised in the original review.

Reviewer #2 (Remarks to the Author):

Dear Authors,

I am satisfied with the additions to the manuscript and replies to my questions.

Reviewer #3 (Remarks to the Author):

satisfactory at all levels

We thank all referees for their positive comments on our revised manuscript.

REVIEWERS' COMMENTS:

Reviewer #1 (Remarks to the Author):

The response to the review satisfactorily addresses the major points raised in the original review.

Reviewer #2 (Remarks to the Author):

Dear Authors,

I am satisfied with the additions to the manuscript and replies to my questions.

Reviewer #3 (Remarks to the Author):

satisfactory at all levels